# Evaluation of Two Vaccines against Foot-and-Mouth Disease Used in Transcaucasian Countries by Small-Scale Immunogenicity Studies Conducted in Georgia, Azerbaijan and Armenia

**DOI:** 10.3390/vaccines12030295

**Published:** 2024-03-12

**Authors:** Efrem Alessandro Foglia, Tengiz Chaligava, Tamilla Aliyeva, Satenik Kharatyan, Vito Tranquillo, Carsten Pötzsch, Cornelis van Maanen, Fabrizio Rosso, Santina Grazioli, Emiliana Brocchi

**Affiliations:** 1Istituto Zooprofilattico Sperimentale della Lombardia e dell’Emilia Romagna (IZSLER), 25124 Brescia, Italy; vito.tranquillo@izsler.it (V.T.);; 2European Commission for the Control of Foot-and-Mouth Disease (EuFMD), FAO, 00100 Rome, Italytamilla.aliyeva@afsa.gov.az (T.A.); satenik.kharatyan@gmail.com (S.K.); cornelis.vanmaanen@fao.org (C.v.M.);; 3National Food Agency (NFA) of the Ministry of Environmental Protection and Agriculture (MEPA), Tbilisi 0159, Georgia; 4Azerbaijan Food Safety Institute (AFSI) at the Azerbaijan Food Safety Agency (AFSA), Baku 1069, Azerbaijan; 5Scientific Center for Risks Assessment and Analysis in Food Safety Area—CJCS of Republic of Armenia, Yerevan 0071, Armenia

**Keywords:** foot-and-mouth disease, small-scale immunogenicity study, vaccination campaign, vaccine effectiveness assessment

## Abstract

In countries endemic for foot-and-mouth disease (FMD), routine or emergency vaccinations are strategic tools to control the infection. According to the WOAH/FAO guidelines, a prior estimation of vaccine effectiveness is recommendable to optimize control programs. This study reports the results of a small-scale immunogenicity study performed in Transcaucasian Countries. Polyvalent vaccines, including FMDV serotypes O, A (two topotypes) and Asia1 from two different manufacturers, were evaluated in Georgia, Azerbaijan and Armenia. Naïve large and small ruminants were vaccinated once and a subgroup received a second booster dose. The titers of neutralizing antibodies in sera collected sequentially up to 180 DPV were determined through the Virus Neutralization Test versus homologous strains. This study led to the estimate that both the vaccines evaluated will not induce a protective and long-lasting population immunity, even after a second vaccination, stressing that consecutive administrations of both vaccines every three months are mandatory if one aspires to achieve protective herd immunity.

## 1. Introduction

Foot-and-mouth disease (FMD) is a serious transboundary infectious disease of cloven-hooved animals, which leads to considerable socio-economic impacts [1,2]. The economic effect of the disease is related to various factors, including a reduction in livestock production, the limitation of affected countries in access to global markets and also costs related to the control of the disease and the gain or regain of FMD free status [1]. The estimation of the annual impact of FMD in 2013 in endemic countries was between USD 6.5 and 21 billion [1]. Currently, FMD circulates endemically in African and Asian continents, with occasional recurrence in South America [3]. The circulation of the disease in North Africa, in the Asian region of Türkiye (Anatolia) and in all the Middle East, together with sporadic incursions across the borders between the Russian Federation and the Republic of Kazakhstan, Mongolia or the People’s Republic of China, makes the disease a continuous threat to countries with a highly developed livestock production industry, where FMD has been eradicated [3,4]. Thus, the prevention and control of the disease is a high priority, strongly influencing animal health policies in many countries.

Transcaucasian Countries (TCC) play a strategic and crucial role in preventing the circulation of FMD virus (FMDV, genus Aphtovirus, family Picornaviridae) in Europe, since the early 1990s. In fact, they border to the north with a zone of the Russian Federation which is recognized by the WOAH (World Organization for Animal Health) as free from FMD with vaccination, and to the south with two endemic countries, namely Türkiye and Iran (https://www.foot-and-mouth.org, accessed on 15 July 2023). Over the years, the EuFMD (European Commission for the control of FMD), in collaboration with the FAO (Food and Agriculture Organization), the WOAH and the European Union, supported activities to reduce the FMD risk together with TCC and neighboring countries, including vaccination campaigns [5]. This led to the recognition of stage 2 of the Progressive Control Pathway for FMD (PCP-FMD; https://www.foot-and-mouth.org, accessed on 15 July 2023) for Armenia and Azerbaijan and of stage 3 for Georgia, with only sporadic and spatially limited circulation of the virus. The most recent report of the disease dates back to 2015 in Armenia (A/ASIA/G-VII) [6].

Indeed, vaccination remains the most powerful prophylactic tool to control FMD outbreaks, to reduce the spread of the virus from endemic countries and to limit its spread after incursions [7]. For this reason, approximately 2.5 billion doses of FMD vaccines are globally administered to livestock animals [1]. Nowadays, many commercial vaccines against FMD are available, but because of the high antigenic variability of the causative agent of the disease, the selection of the most effective vaccine formula is highly dependent on the area where the vaccine will be used [8]. In fact, one exciting challenge in the fight against FMD is that the virus exists as seven serotypes (O, A, C, Asia1, and Southern African Territory [SAT] 1–3) and is subject to a constant evolutionary pressure, because of its small single-stranded RNA genome, which results in a continuous emergence of new variants [2,9,10,11]. Therefore, the inter- and intra-serotype antigenic differences make properly selecting the virus strains to be included in vaccines imperative, since adequate antigenic matching between circulating FMDVs and vaccine strains is the basis of the effectiveness of a vaccination program. Some other factors limit the effectiveness of FMD vaccines, such as the relatively short duration of induced immunity and the restrictive requirement to maintain a stable cold chain [12] due to the lability of the FMD virions. The implementation of small-scale vaccination trials is a useful means to evaluate the actual efficacy of FMD vaccines in conferring immune protection to the vaccinated animals [13] and thereafter to predict the effectiveness of FMD vaccination campaigns [14,15,16], especially if manufacturers do not provide all that information crucial to properly evaluate vaccine effectiveness, in particular against the circulating lineages [17]. The FAO and the WOAH, conscious of the importance of the vaccines in control of FMD programs, have recently published specific guidelines for vaccination and post-vaccination monitoring in order to assess the effectiveness of the FMD vaccination campaign [17]. The present study describes the results of a small-scale immunogenicity studies (SSIS) program promoted by the EuFMD, performed as part of the activities aimed to support the continuous evaluation of the performance of the FMD vaccination campaigns in Georgia, Azerbaijan and Armenia, and refers to data collected between 2018 and 2019. The scheme of these SSIS performed in TCC countries followed the guidelines of the FAO and the WOAH about vaccination for FMDV and post-vaccination monitoring [17]. The immunogenicity studies carried out were focused on the measurement of antibodies neutralizing FMDV strains phylogenetically close to those included in the administered vaccines. They provided empirical data estimating the immune response induced by the vaccination in host species, useful to assess vaccine effectiveness, comparing the obtained data with those from previous studies that, combined with the veterinarians’ expertise gained in the field, indicated that vaccination with an effective vaccine against FMD should induce seroconversion in at least 70% to 80% of the population [18]. The trials followed a serological approach, mainly based on the Virus Neutralization Test (VNT), since given the relationship demonstrated between neutralizing antibody titers and probability of protection [18], it enables a more reliable evaluation of the effectiveness of the vaccines [17,19,20].

## 2. Materials and Methods

### 2.1. Ethical Statement

This study has been carried out in accordance with national and international guidelines, ensuring complete adherence to best practices of veterinary care. National centers entered in a verbal agreement with farmers or stakeholders to vaccinate animals and collect samples.

### 2.2. Animals

According to the experimental design suggested by the FAO/WOAH for small-scale trials [17], 40 animals in each of the three countries were to be enrolled with the following prerequisites: aged between 6 and 12 months, never been exposed to FMDV infection, reportedly non-vaccinated, seronegative for antibodies against non-structural proteins of FMDV (FMDV-NSP) and without maternal antibodies derived from vaccinated or infected mothers. The main group included 20 large ruminants (LR; cattle) and 20 small ruminants (SR; sheep and/or goats). Each subgroup was divided into three sets: 9 animals received a single dose of vaccine at day 0 (LR-01 and SR-01); 9 animals received a first dose of vaccine at day 0 and a second dose at day 90 (LR-02 and SR-02); 2 control animals remained unvaccinated (LR-03 and SR-03).

In Azerbaijan and Armenia, LR and SR were recruited and kept in a unique farm (one for each species of animals), whilst few close farms were involved, all situated in the same area, in Georgia.

### 2.3. Vaccines

Two trivalent vaccines were used in this study. The first was produced by Shchelkovo Biocombinat (Ščëlkovo, Moscow Oblast, Russian Federation) and administered in Georgia and Azerbaijan. In Armenia, animals received the vaccine manufactured at the Federal Center for Animal Health FGBI “ARRIAH” (Vladimir, Vladimir Oblast, Russian Federation). Both vaccines contained the following virus lineages: O/ME-SA/PanAsia2, A/ASIA/Iran-05, A/ASIA/G-VII and Asia-1/ASIA (lineage Shamir for the Shchelkovo Biocombinat vaccine and lineage Sindh8 for the ARRIAH vaccine), providing at least 6 PD50 for each valency, according to manufacturers’ information. The vaccines were produced starting from inactivated FMDV strains, cultured on a suspension of BHK-21 cell line (ATCC CCL-10), and contained aluminum hydroxide as sorbent and saponin as adjuvant. Shelf life was 18 months and storage at a temperature of 2–8 °C is essential for preventing antigens degradation. As for manufacturers’ instructions, each vaccine dose was of 2 mL and 1 mL for LR and SR, respectively. For development and maintenance of adequate immunity, manufacturers suggest vaccinating animals every 3 months from the age of 3–4 months until 18 months, then every 6 months.

### 2.4. Sampling Scheme

Blood samples were collected following an agreed scheme and were taken by venipuncture from the jugular vein. The scheme included the following samplings:-Before vaccination (0 DPV);-14 days post-vaccination (14 DPV), for evaluation of early immune response to vaccine;-28 days post-vaccination (28 DPV), as standard time to evaluate vaccine-induced immune response;-60 days post-vaccination (60 DPV);-90 days post-vaccination (90 DPV), to evaluate the duration of antibodies; at this time, one half of the animals received a second vaccination;-120 days post-vaccination (120 DPV), corresponding to 30 days after the second vaccine administration, to quantify the immune response to a booster vaccination;-150 days post-vaccination (150 DPV), corresponding to 60 days after the second vaccination;-180 days post-vaccination (180 DPV), i.e., 90 days after the second vaccination.

Blood samples were centrifuged at 3000× *g* for 6 min and two aliquots of 2 mL for each serum were stored at −20 °C.

### 2.5. Discrepancies with the Original Design

Because of logistic reasons related to the field studies, some variations to the original design were introduced: in Azerbaijan, 14 cattle (instead of 9) were enrolled in the group LR-01 and 4 in the unvaccinated group (LR-03); concerning SR, only the group of 9 SR-01, receiving a single vaccination, has been included in the trial. Both in Georgia and Armenia the group LR-01 was composed of 10 instead of 9 cattle (Figure 1).

Major divergencies were found in the samplings’ availability, as graphically represented in Figure 1. In particular, in Georgia and Azerbaijan, the number of LR and SR decreased over time because of natural death, slaughtering or other situations. In Armenia, sampling of LR after the booster vaccination was not performed, while the tubes of SR lost their tags and were not processed. In Azerbaijan the last samplings were carried out at 120 DPV.

### 2.6. ELISA for the Detection of Antibodies against NSP-FMDV

Screening was performed using the 3ABC trapping indirect ELISA developed and extensively validated at the IZSLER laboratory [20,21] and available as a ready-to-use kit (FMDV 3ABC-trapping ELISA, IZSLER, Brescia, Italy), where sera are tested at 1/100 dilution against the 3ABC recombinant antigen immune-captured by a monoclonal antibody. A peroxidase-conjugated monoclonal antibody specific for ruminant IgG is used as detector system. Results are expressed as percentage positivity of test sera referring to the reaction of a strong positive control serum present in each ELISA plate, used as 100%. Test sera are considered positive if percentage positivity is ≥10%.

### 2.7. The Virus Neutralization Test (VNT)

The VNT is the gold-standard method used to evaluate the effectiveness of FMD vaccines as antibodies able to neutralize virus infectivity in vitro are supposed to be better correlated with protection [17,20,21]. VNTs were carried out by testing in duplicate sequential dilutions of sera, starting from 1/16 (final serum dilution in the serum/virus mixture), against a virus dose of 100 Median Tissue Culture Infectious Doses (TCID_50_) of virus using IB-RS-2 cell line according to the method outlined in the WOAH Manual [20] and certified as ISO17025 [22] in IZSLER lab. The strains used for the test belong to the same lineages to those included in the vaccine and were recently circulated in areas neighboring TCC, strongly suggesting a plausible antigenic correlation. In detail, the strains, provided by Şap Institute (Ankara, Türkiye), were O/ME-SA/PanAsia2/TUR/07; A/ASIA/Iran-05/TUR/06; A/ASIA/G-VII/NEP/84; Asia1/ASIA/Sindh8/TUR/15.

VNT titers were expressed as log_10_ of the reciprocal of the highest dilution protecting 50% of the inoculated cell cultures. The adopted threshold for positivity, in line with the WOAH Manual, is 1/32 (log_10_32 = 1.5); therefore, titers > 32 (final serum dilution in the serum/virus mixture) are considered positive [20].

### 2.8. Statistical Analysis

Due to the reported discrepancies from the original plan (Figure 1 and Section 2.5) and in order to evaluate the most suitable test to apply for statistical analysis, a preventive normality test (in detail, the Shapiro–Wilk test [23]) was performed. Because the data distribution was not normal, and groups were considered as independent (statistical hypothesis), the Wilcoxon–Mann–Whitney T test was used for paired non-parametric continuous data on medians. *p*-values < 0.05 were considered statistically significant [24]. Reported *p*-values refer only to noteworthy pairs of data (e.g., data collected before and after vaccine administrations). Because the two subgroups (01 and 02) before 90 DPV received the very same treatment (one dose of vaccine) and did not show statistical differences, they were analyzed as a single group from 0 DPV to 90 DPV.

The point and interval estimate of the proportion of animals that converted on the different days of observation was obtained by the Agresti–Coull method [25], using the binom.confint() function of the binom package (1.1-1.1) [26] and the results are shown in as a graph using functions from the ggplot2 package (3.2.2) [27] in the R environment (4.2.3) [28].

## 3. Results

The guidelines related to the small-scale trials for the evaluation of vaccine quality envisage the involvement of animals from six to nine months, in order to exclude maternally derived antibodies. Because some animals in this SSIS, especially cattle in Georgia and Azerbaijan, showed VNT positivity against some of the lineages involved in the studies, they were presumably derived from residual maternal antibodies. The presence of passive immunity might have adversely affected the vaccine-induced immunity, so the data obtained from animals VNT positive for a specific lineage at day 0 were excluded by the analysis. The exclusion included the results of sera sampled from 0 to 180 DPV, associated with the specific virus lineage(s) conferring VNT positivity at day 0. The final number of sera involved in this study is summarized in Table 1.

### 3.1. Large Ruminants (LR)

The absence of clinical signs consistent with FMD in the animals involved in the studies, combined with the negative results of NSP-ELISA, carried out on all samplings of sera, confirmed that no incursion of FMDV took place during the SSIS. Moreover, all the animals involved in this study were naïve (negative VNT results; Appendix A). The immune response induced by vaccination in LR was estimated in the three involved countries with minor differences in the study scheme. In detail, testing of sera available from Georgia and Azerbaijan provided information about primary immune response and booster effect after Shchelkovo Biocombinat vaccine administration and, thanks to later samplings performed in Georgia (150 and 180 DPV), also duration of the secondary immunization was assessed. Serum samplings available from Armenia enabled the evaluation of only the primary immunization induced by a single-dose administration of the ARRIAH vaccine.

All sera sampled by cattle belonging to the control group (LR-03), that did not receive any vaccination, were tested for NSP protein and with the VNT test and were confirmed negative for all the duration of the trial in the three involved countries, corroborating again that no incursion of the virus could be detected during the SSIS period.

#### 3.1.1. LR—Serotype O

Vaccinated cattle reacted differently, according to the received vaccine. Overall, a single-dose administration of the Shchelkovo Biocombinat vaccine induced an inadequate immune response (Figure 2). Fourteen days after the administration only few animals, precisely 31% in Georgia and 33% in Azerbaijan showed positive VNT titers (Appendix A). Nevertheless, the medians of titers remained under the positivity threshold in both countries (Figure 2), pointing out that the immune response induced by the administration of a single dose of this vaccine was likely not sufficient to effectively induce immunization against serotype O. Over time, medians and positivity percentage remained stable, especially in Azerbaijan (Figure 2b), or at least decreased, such as in Georgia (Figure 2a; Appendix A).

The single-dose administration of the ARRIAH vaccine used in the Armenia study induced a significative immunization (Figure 2c; *p* value ≤ 0.001), if compared to the immune response observed against the Shchelkovo Biocombinat vaccine in the other two countries. At 14 DPV all the collected sera showed VNT titers over the positivity threshold with peak values of 2.28 log_10_ (Figure 2c, Appendix A); however, the antibody titers quickly decreased in the following samplings and after three months the medians of VNT titers settled across the positivity threshold (Figure 2c), confirming that immunity against serotype O induced by administration of one dose of the ARRIAH vaccine was effective but not long-lasting.

The administration of a second dose, performed at 90 DPV and assessable only for the Shchelkovo Biocombinat vaccine (Figure 2a,b), induced a significant booster immunization against serotype O in cattle, clearly detectable one month after the second vaccination (120 DPV in Georgia and Azerbaijan; *p* values ≤ 0.05) and persisting in following samplings, even though 180 DPV (as shown in the Georgian study) medians of VNT titers had decreased, but still remaining well above the positivity threshold (Figure 2a).

#### 3.1.2. LR—Serotype A

Both the used vaccines contained two serotype A strains, belonging to lineages A/ASIA/Iran-05 and A/ASIA/G-VII. The immune response against the two A lineages in cattle vaccinated with the Shchelkovo Biocombinat vaccine (trials performed in Georgia and Azerbaijan) was not homogeneous (Figure 3). Overall, the immune response against the first vaccine dose of A/Iran-05 was poorer in both countries (Figure 3a,b). In Georgia, the median of VNT titers did not reach the positivity threshold (Figure 3a; *p* value ≤ 0.01) and only 33% of sera resulted positive at 14 DPV (Appendix A). In Azerbaijan, 68% of cattle were able to neutralize A/Iran-05 virus after receiving a single dose of vaccine (Appendix A), with a median of VNT titers 1.86, but comparing the medians of VNT titers before and after vaccination, only small significative differences were detected (Figure 3b; *p* value ≤ 0.05). A significant change in immune status was detected in groups LR-02, which received a second dose of vaccine (Figure 3a,b). One month after the re-vaccination (120 DPV), VNT titers against A/Iran-05 increased, reaching very high values, with medians approaching or even higher than 3 log_10_ (Georgia ≃ 2.81, Azerbaijan ≃ 3.29; *p* values ≤ 0.01). In the Georgian trial, antibodies decreased rapidly; and 3 months after the second vaccination, two sera were negative, while the median of VNT titers was log_10_ 1.81 (Figure 3a and Appendix A).

Immune response against the component A/G-VII present in the Shchelkovo Biocombinat vaccine was better than against the component A/Iran-05. Overall, a single administration of vaccine induced the arise of neutralizing antibodies (*p* value ≤ 0.001 for Georgia and *p* value ≤ 0.01 for Azerbaijan) and the majority of the bovine sera collected in Georgia and Azerbaijan after vaccination were tested positive to the VNT at 14 DPV (Figure 3d,e). The medians of VNT titers remained above the positive threshold over time (Figure 3d,e). The second administration of the Shchelkovo Biocombinat vaccine induced a booster effect in the studies conducted in both countries, even if it was statistically significative only for Georgia (*p* value ≤ 0.01). The medians of VNT titers against A/G-VII at 120 DPV reached high values (3.41–3.59) in samples collected in both countries (Figure 3d,e). Finally, SSIS conducted in Georgia revealed that the immunization induced by two vaccine administrations persisted for 3 months after the re-vaccination (180 DPV), though with decreasing antibody titers, and only one serum returned a negative result to the VNT against A/G-VII (Figure 3d and Appendix A).

Administration of the ARRIAH vaccine in Armenia induced a faster response against A/Iran-05 with positive VNT titers already detectable at 14 DPV. Nevertheless, when reaching the peak (28 DPV) both lineages induced a comparable immunization in terms of percentage of positivity (88% A/Iran-05 and 81% A/G-VII, Appendix A). Over time, the medians of VNT titers remained nearly constant above the positivity threshold, but 3 months after vaccination almost all tested sera were negative (Figure 3c,f and Appendix A).

#### 3.1.3. LR—Serotype Asia1

Data on the first immunization of LR against serotype Asia1, obtained in Georgia after administration of the Shchelkovo Biocombinat vaccine, did not reveal any new trend (Figure 4a) when compared to the observed immunization against serotype O (Figure 2a) and against the lineage A/Iran-05 (Figure 3a). In detail, one half of the cattle population reacted positively to vaccination at 14 DPV (Appendix A) but the median of VNT titers never crossed the positivity threshold (Figure 4a), pointing out that the immune response was inadequate after the administration of one single dose in naïve cattle. The neutralizing abilities of collected sera against serotype Asia1 significantly increased after the second vaccination (Figure 4a and Appendix A; *p* value ≤ 0.01). At 120 DPV a significant booster effect was detected, with the median of VNT titers at 2.7 (Figure 4a). However, as observed for serotypes O and A, the medians of VNT titers decreased in the following two months (Figure 4a). In Azerbaijan (Figure 4b), the primary antibody response induced by the Shchelkovo Biocombinat vaccine was higher than that detected in Georgia (Figure 4a; *p* value ≤ 0.001): the majority of samplings showed positivity to the VNT against Asia1 at 14 DPV (Appendix A) and the medians of VNT titers, from log10 1.68 declined rapidly in the subsequent sampling until 90 DPV (Figure 4b), when only 38% of sera was still VNT positive. The second vaccine dose induced the expected booster effect, though the median of VNT titers (2.16) remained lower than that observed in Georgia (Figure 4b; *p* value ≤ 0.05).

The ARRIAH vaccine, used in the Armenia trial, induced a poor immune response against serotype Asia1 with a single administration (Figure 4c). At 28 DPV, the majority of the population involved in the SSIS showed seroconversion (Appendix A) but the median of titers just boarded the positivity threshold (Figure 4c), declining quickly to negative (Figure 4c).

### 3.2. Small Ruminants (SR)

As for LR (3.1), data obtained from NSP-ELISA on SR sera, together with the absence of FMD clinical signs, excluded the circulation of FMDV during the whole study. The evaluation of a vaccination-induced immunization against FMDV in SR was performed only on sera collected in Georgia and Azerbaijan. Georgian trial allowed to assess both primary and secondary immune response induced by the Shchelkovo Biocombinat vaccine, while only one single dose was administered to SR in Azerbaijan. Unfortunately, SR sera from Armenia were unsuitable to be analyzed (Figure 1).

In the studies conducted on small ruminants, all sera sampled from control group animals (SR-03) were negative according to NSP-ELISA and the VNT.

#### 3.2.1. SR—Serotype O

Vaccination of SR in Georgia with a single dose induced a good but short immunization against serotype O (Figure 5). At 14 DPV the median of VNT titers was 1.86 (*p* value ≤ 0.001) and almost all animals were positive to the VNT, but at 90 DPV median was under the positivity threshold (Figure 5a and Appendix A). The second vaccination induced a significative booster effect, which was comparable to the primary immune response (Figure 5a). The median of VNT titers in double-vaccinated animals at 120 DPV was 2.2 log_10_ (*p* value ≤ 0.05) and sampling at day 180 still showed a median over the positivity threshold (Figure 5a). In Azerbaijan, the administration of a single dose of vaccine induced seroconversion in the majority of the SR population involved in the SSIS (80%, Appendix A), with median VNT titers of 2.16 log_10_ at 14 DPV, but gradually decreasing in the following samplings (Figure 5b and Appendix A).

#### 3.2.2. SR—Serotype A

Immune response against serotype A in SR vaccinated in Georgia (Figure 6a–c) was comparable to that previously described for serotype O (Figure 5a). Briefly, 14 days after the first vaccination, nearly all animals showed a growth in neutralizing antibodies against both serotype A lineages (Appendix A), with medians of VNT titers against A/Iran-05 at 2.11 log_10_ and 2.41 log_10_ against A/G-VII (Figure 6a,c; *p* values ≤ 0.001), decreasing over time. Medians were negative starting from sampling at 60 DPV for A/Iran-05 (Figure 6a) and at 90 DPV for A/G-VII (Figure 6c). Interestingly, the decline in VNT titers was faster for A/Iran-05 than for A/G-VII, resulting in the percentage of A/G-VII-positive animals remaining over 80% until 60 DPV, versus 50% against A/Iran-05 (Appendix A). The secondary immunization against A/Iran-05, induced by revaccination, involved all animals of the SR-02 group and was comparable to the primary response both in terms of magnitude and duration (Figure 6a and Appendix A). Medians of VNT titers of primary (14 DPV) and secondary (120 DPV) responses were similar, 2.11 and 2.16, respectively, with minimum serum collected 3 months after the administration of the vaccine (Figure 6a). The secondary response against A/G-VII elicited a significant booster of neutralizing antibodies, with the highest median of log_10_ VNT titers of 3.02 at 120 DPV, but again of short duration (Figure 6c; *p* value ≤ 0.01). In Azerbaijan, the administration of the same vaccine (Shchelkovo Biocombinat) induced a more durable immunization over time (Figure 6b,d), with a peak at 28 DPV against A/Iran-05, when sera sampled reached a median of log_10_ VNT titers 2.01 for neutralizing antibodies (Figure 6b), while sera reached the highest median of log_10_ VNT titers (2.48) at 60 DPV against A/G-VII (Figure 6d).

#### 3.2.3. SR—Serotype Asia1

In Georgia, vaccination of SR with the Shchelkovo Biocombinat vaccine induced a trend in the antibody response against serotype Asia1 (Figure 7a) comparable to those observed against serotype O (Figure 5a) and A (Figure 6a,c). Both administrations of vaccine (first and second) stimulated immunization covering at least the 75% of the population (Appendix A): at the first samplings after each administration, 14 DPV and 120 DPV, peaks of neutralizing antibodies were detected with medians of log_10_ 2.11 (*p* value ≤ 0.01) and log_10_ 2.31 (*p* value ≤ 0.05), respectively (Figure 7a). Sera sampled three months after each administration (90 DPV and 180 DPV) were VNT negative (Figure 7a), confirming that the duration of immunization never exceeded that period. In the Azerbaijan trial, where SR received a single dose of the Shchelkovo Biocombinat vaccine, seroconversion was detected from 14 DPV in 71% of animals (Appendix A), reaching a median of VNT titers of log_10_ 1.86, with a minor fluctuation in the following sampling at 28 DPV and a tendency to decrease in subsequent samplings (Figure 7b).

### 3.3. Comparison between the Observed and the Expected Proportion of the Immunized Population

The analysis of the 95% confidence interval and point estimate of proportion animals with a positive VNT titer highlighted the global situation related to immunization of LR and SR after vaccination (Figure 8).

The vaccine strain inducing the highest immunization with a single vaccination was the A/G-VII (Figure 8a). All the involved animal groups showed a percentage of positivity to the VNT in or over the expected range (70–80%) between 14 and 28 DPV; this situation is also persistent in Georgian SR until 60 DPV and in Azerbaijani animals (both LR and SR) until 120 DPV (Figure 8a). The other serotype A strain, A/Iran-05, induced an effective immunization (proportion of the population positive to the VNT equal or higher than 70%) with a single dose only in Armenian LR and Georgian SR until 28 DPV, while in all the other groups, the percentage of VNT-positive animals after a single vaccination was lower than expected (Figure 8b). The serotype Asia1 included in the vaccine formula, induced an overall poor immunization level of the population after a single-dose administration. Only Azerbaijani animals (both LR and SR) and Georgian SR reached the expected percentage at day 14, but the immunization was not long lasting. Further, in Azerbaijan, the vaccinated SR and LR populations showed a VNT positivity rate between 70% and 80% also at 120 DPV (Figure 8c). Finally, serotype O showed a good immunization rate after one vaccination, higher than expected, only in Armenian LR, between 14 DPV and 60 DPV, and in SR of Azerbaijan and Georgia at day 14 (Figure 8d).

However, results for the Georgian animals who received a second dose of vaccine differed. Double-vaccinated LR proved a good immunization level (VNT positivity) against all the tested FMDV lineages in more than 80% of the population at day 120 and the immunity rate of the LR population remained over the expected threshold (75–80%) until 180 DPV. Conversely, the SR population, despite a clear reaction to the second dose administration, did not reach the expected rate of VNT positivity induced by the booster vaccination, except for the A/G-VII strain (Figure 8).

## 4. Discussion

The small-scale immunogenicity studies (SSIS) performed and described in this paper aimed to determine the immunogenic responses in large and small ruminants induced by selected FMD vaccines, already administered to FMD-susceptible species in Transcaucasian Countries (TCC). In detail, this SSIS was finalized to the evaluation of two similar vaccines, manufactured by two companies (Shchelkovo Biocombinat and ARRIAH), and containing the same four component vaccine lineages: one for serotype O, two for serotype A, and one for serotype Asia1 (Section 2.3). Similar vaccines have been the object of studies in other countries and with analogous aims [16,29,30]. Employing field isolates representing strains phylogenetically closely related to those included in the vaccines’ formula and circulating in neighborhood countries (namely in Türkiye), the data obtained during this SSIS also provided an evaluation of the immunological response against some of the FMDV strains circulating in the Middle East area and thus posing great epidemiological risks to TCC. The latest FMDV lineages which circulated in TCC and in their neighboring countries were exactly those involved in this work: O/ME-SA/PanAsia2 (Georgia 2011, Türkiye 2022 and Iran 2023), A/ASIA/Iran-05 (Iran 2021), A/ASIA/G-VII (Türkiye 2020 and Armenia 2015) and Asia1/ASIA/Sindh8 (Iran 2020) as reported by the World Reference Laboratory for Foot-and-Mouth Disease (wrlfmd.org, accessed on 15 July 2023).

In this work, the effectiveness of vaccination was assessed using the VNT to evaluate serological response in treated animals, subsequent to an explorative NSP-ELISA test to exclude any prior FMDV infections and/or the circulation of virus in the experimental population. Titers obtained by the VNT were always considered only as indicators of the magnitude of the immune response induced by the vaccine and of its duration. No speculations were made about the correlation between neutralizing antibody presence and the protection level of animals from FMDV infection.

Overall, the data obtained by the VNT on sera of vaccinated animals provided many and various information. But to thoroughly analyze the immunological response induced by the vaccination, it is important to consider two limitations noticed during the analysis of obtained data. First, neutralizing antibodies against the lineages O/ME-SA/PanAsia2 and A/ASIA/Iran-05 were detected before vaccination in some Georgian and Azerbaijani cattle. These results may be justified with maternal antibodies which might have inhibited the vaccine-induced immunity. Though there are no evident data validating this hypothesis, we excluded those results from the analysis in order to avoid any influence on the actual evaluation of effectiveness of the two vaccines. Second, every single group showed variability between data, highlighting large ranges in titers. The observed distribution of results could be related to many factors, including natural intrinsic variability in the immune response among different animals, heterogeneous level of preservation of the vaccine during the campaign (especially related to the maintenance of the cold chain) and difficulty in vaccinating the animals and efficacy of the actions (e.g., sheep wool may hinder needle penetration). Excluding the results related to the potential maternally protected animals, the obtained data could be interpreted following three comparison schemes: (i) the FMDV lineage, (ii) the administered vaccine and (iii) the species of vaccinated animals.

(i) Considering all the collected data, the most immunogenic between the lineages included in the vaccines is the A/ASIA/G-VII (Figure 3, Figure 6 and Figure 8) possibly due to higher antigen stability or a higher concentration. This lineage induced a good immune response in vaccinated animals, already 14 days after the first administration, with the medians of log_10_ VNT titers between 1.38 and 2.41, affecting > 70% of the population. Furthermore, it showed the most efficient booster effect after the second vaccination, performed at 90 DPV, with medians of VNT titers between log_10_ 3.02 and 3.59, involving > 80% of the LR population and with long-lasting persistence of neutralizing antibodies in the animal sera.

The other three lineages stimulated poorer immunization, compared to A/ASIA/G-VII. First vaccinations induced ineffective immune response with low or negative medians log_10_ VNT titers at 14 DPV (between 1.20 and 2.16) except for Armenian cattle against type O with a median log_10_ VNT titer of 2.2. Those titers decayed very quickly, and between 28 and 60 DPV almost all medians were below the VNT positivity threshold. The second vaccination induced significative booster effects in >70% of the population, according to expectations, with a median of log_10_ VNT titers between 2.16 and 3.29.

Analyzing the duration of neutralizing antibodies, all four lineages showed almost the same pattern: antibodies after first vaccination were not persistent in vaccinated animals, their concentration decayed quickly and often the duration of the effects of immunization were undetectable three months after the administrations. The immunity induced by the second administration of vaccines showed better VNT titers over time, even if the detected decay is to assume that the immunization would not remain after three months. These observations are indeed consistent with indications of both vaccines manufacturers, which suggest a vaccination regimen based on consecutive administration every three months until 18 months of age.

(ii) Comparing the results obtained vaccinating ruminants with the Shchelkovo Biocombinat vaccine (Georgia and Azerbaijan) and the ARRIAH vaccine (Armenia) some differences in immune response are clearly detectable. The comparison of the vaccines was performed considering only the first immunization of LR.

The ARRIAH vaccine induced better immunization in animals with a single vaccine dose. In particular, for lineage O/ME-SA/PanAsia2, ARRIAH elicited the highest response of 14 DPV in >80% of the population, with a median of log_10_ VNT titers of 2.28, while the Shchelkovo Biocombinat vaccine induced immunization with medians of log_10_ VNT titers between 1.20 and 1.38 (Figure 2 and Figure 8). For the other three lineages, after vaccination with the ARRIAH vaccine, animals reached the highest concentration of neutralizing antibodies at 28 DPV (medians of log_10_ VNT titers 1.51–1.89) with titers greater than those observed after vaccination with Shchelkovo Biocombinat and involving a greater part of the population (Figure 3, Figure 4, Figure 6, Figure 7 and Figure 8). Single-vaccinated LR from Georgia and Azerbaijan reached a median of log_10_ VNT titers lower than 1.81 between 14 and 28 DPV. The only exception was the immune response of Azerbaijani cattle against lineage A/ASIA/G-VII, which was detected in >80% of the population and was characterized by a median of log_10_ VNT titers between 1.98 and 2.35 (Figure 3 and Figure 8). Unfortunately, without data available during the secondary immune response (120 and 150 DPV), the booster effect of the ARRIAH vaccine could not be evaluated or compared with the good one elicited by the Shchelkovo Biocombinat vaccine. Overall, despite the differences in the trend and level of immune response induced by the two vaccines, both demonstrated lower efficiency than expected, due to the modest length of response (less than three months), to the low VNT titers immediately after the administration (14–28 days) and the rate of the VNT-positive population.

(iii) Comparison between data obtained testing sera from LR and SR showed an evident difference in the trend of immune response of cattle and small ruminants. This evaluation has been performed considering data collected in Georgia.

LR showed a general ineffective immune response, in terms of VNT titers and rate of the positive population after a single-dose vaccination (median of log_10_ VNT titers between 1.20–1.68), but the second dose of vaccine elicited a significant booster effect (2.16–3.41; positive rate >80% of LR). Conversely, the SR immune system reacted more intensely to the first vaccination and at 14 DPV, higher medians of log_10_ VNT titers against all four lineages, ranging between 1.86 and 2.41, were detected. Interestingly, the booster effect in SR, albeit clearly detected, was not proportionate to the first immunization nor comparable to the trend observed in cattle with the medians of log_10_ VNT titers growing only to 2.16–3.02. In other words, LR showed a second immunization 1–2 log_10_ higher than the first, while SR showed a booster effect of only ~0.5 log_10_ magnitude. In addition, the rate of the VNT-positive SR population overcame the expected threshold only after the first immunization and not after the second (Figure 8). Immune response was demonstrated to be short lived in both species: in fact, almost all animals were negative within three months after vaccine administration.

These data, together with the results of previous works [16,21], confirm the high value of performing SSIS to support FMD vaccine campaigns and reinforce the importance of routine FMD vaccination in endemic countries to elicit adequate antibodies’ titers in the vaccinated animals and to ensure that the vaccination fits, as well as possible, to the specific situation of the area involved in the campaign, in terms of circulating strains, susceptible species to be vaccinated and timing of vaccination.

Finally, the present work also highlights the difficulties related to SSIS, especially those correlated with the work conducted in the field. The possibility to reduce the variability, to follow more rigorously the design and to work in a more controlled environment may increase the robustness of this kind of studies. However, despite some limitations and deviations from the original design of the studies described, the data collected highlighted useful results for the intended purpose, defining the limits of the vaccines evaluated and the need for close vaccination boosters to induce, and maintain, an adequate level of antibodies.

## 5. Conclusions

The reported evaluation of FMD vaccines led to the following conclusions. (i) The FMD vaccines involved in these SSIS in TCC elicited poor antibody responses after primary vaccination in terms of antibody titers and the proportion of the immunized population. (ii) Animals receiving a second dose showed a significant increase in the immune response, highlighting that a second dose administration is crucial to enhance the effectiveness of the vaccination. A reduction in the time period between the first and seconds dose to enhance the immune response could be evaluated for both vaccines. (iii) Consecutive vaccinations are needed to improve protection against FMD because the duration of the humoral response induced by the studied vaccines in young, naïve animals was short and generally no longer than three months.

## Figures and Tables

**Figure 1 vaccines-12-00295-f001:**
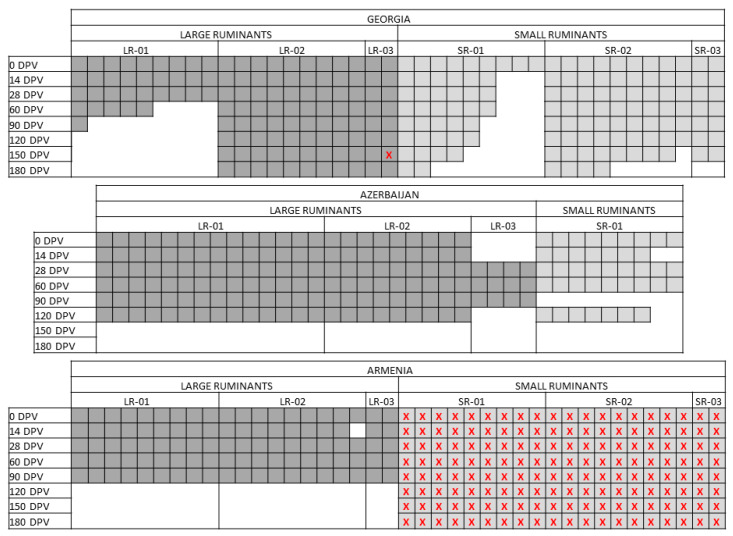
Sampling scheme. The picture represents the actual sampling carried out in the three countries, namely Georgia, Azerbaijan and Armenia. Every square corresponds to a single animal. In dark grey, the large ruminants (LR); in light grey, the small ruminants (SR). Red crosses highlight those samples that could not be analyzed because of empty vials or lost tags.

**Figure 2 vaccines-12-00295-f002:**
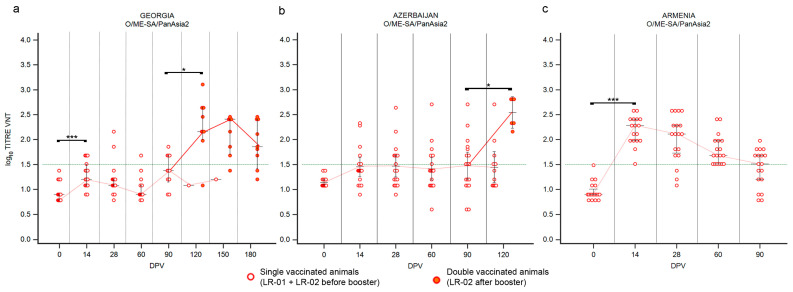
Immune response of large ruminants (LR) against FMDV serotype O. Charts showing the medians of log_10_ VNT titers obtained testing the neutralizing ability of LR sera against FMDV strain O/ME-SA/PanAsia-2/TUR/07. Sera from Georgia (**a**), Azerbaijan (**b**) and Armenia (**c**). Abscissa, days post-vaccination (DPV); ordinate, log_10_ VNT titers. Error bar, 95% confidence interval; VNT positivity threshold, >1.5 log_10_ titer in green dashed line. Empty dots and dashed red line, single-vaccinated animals (LR-01 and LR-02 before booster); filled dots and solid red line, double-vaccinated animals (LR-02 group after booster). ***, *p* value ≤ 0.001; *, *p* value ≤ 0.05.

**Figure 3 vaccines-12-00295-f003:**
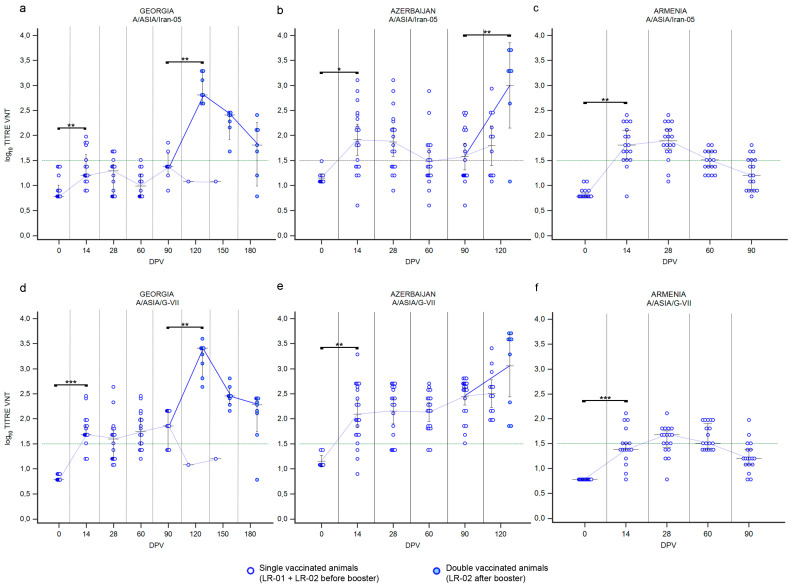
Immune response of large ruminants (LR) against FMDV serotype A. Charts showing the medians of log_10_ VNT titers obtained testing the neutralizing ability of LR sera against FMDV strain A/ASIA/Iran-05/TUR/06 (**a**–**c**) and FMDV strain A/ASIA/G-VII/NEP/84 (**d**–**f**). Sera from Georgia (**a**,**d**), Azerbaijan (**b**,**e**) and Armenia (**c**,**f**). Abscissa, days post-vaccination (DPV); ordinate, log_10_ VNT titers. Error bar, 95% confidence interval; VNT positivity threshold, >1.5 log10 titer in green dashed line. Empty dots and dashed blue line, single-vaccinated animals (LR-01 and LR-02 before booster); filled dots and solid blue line, double-vaccinated animals (LR-02 group after booster). ***, *p* value ≤ 0.001; **, *p* value ≤ 0.01; *, *p* value ≤ 0.05.

**Figure 4 vaccines-12-00295-f004:**
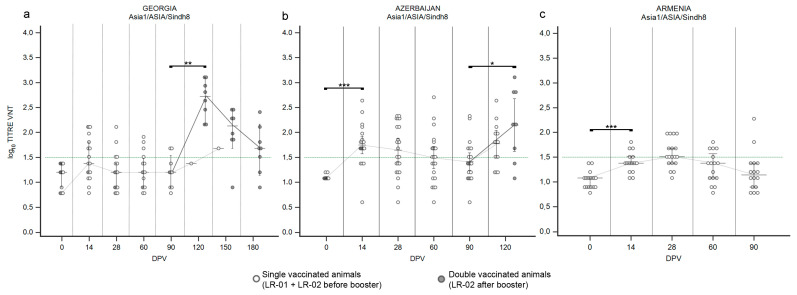
Immune response of large ruminants (LR) against FMDV serotype Asia1. Charts showing the median, maximum and minimum log_10_ VNT titers obtained testing the neutralizing ability of LR sera against FMDV strain Asia1/ASIA/Sindh08/TUR/15. Sera from Georgia (**a**), Azerbaijan (**b**) and Armenia (**c**). Abscissa, days post-vaccination (DPV); ordinate, log_10_ VNT titers. Error bar, 95% confidence interval; VNT positivity threshold, >1.5 log10 titer in green dashed line. Empty dots and dashed grey line single-vaccinated animals (LR-01 and LR-02 before booster); filled dots and solid grey line, double-vaccinated animals (LR-02 group after booster). ***, *p* value ≤ 0.001; **, *p* value ≤ 0.01; *, *p* value ≤ 0.05.

**Figure 5 vaccines-12-00295-f005:**
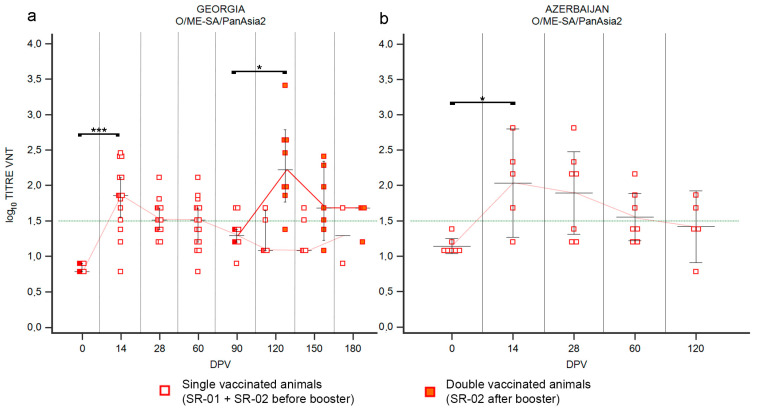
Immune response of small ruminants (SR) against FMDV serotype O. Charts showing the median, maximum and minimum log_10_ VNT titers obtained testing the neutralizing ability of SR sera against FMDV strain O/ME-SA/PanAsia-2/TUR/07. Sera from Georgia (**a**) and Azerbaijan (**b**). Abscissa, days post-vaccination (DPV); ordinate, log_10_ VNT titers. Error bar, 95% confidence interval; VNT positivity threshold, >1.5 log10 titer in green dashed line. Empty squares and dashed red line, single-vaccinated animals (SR-01 and SR-02 before booster); filled squares and solid red line, double-vaccinated animals (SR-02 group after booster). ***, *p* value ≤ 0.001; *, *p* value ≤ 0.05.

**Figure 6 vaccines-12-00295-f006:**
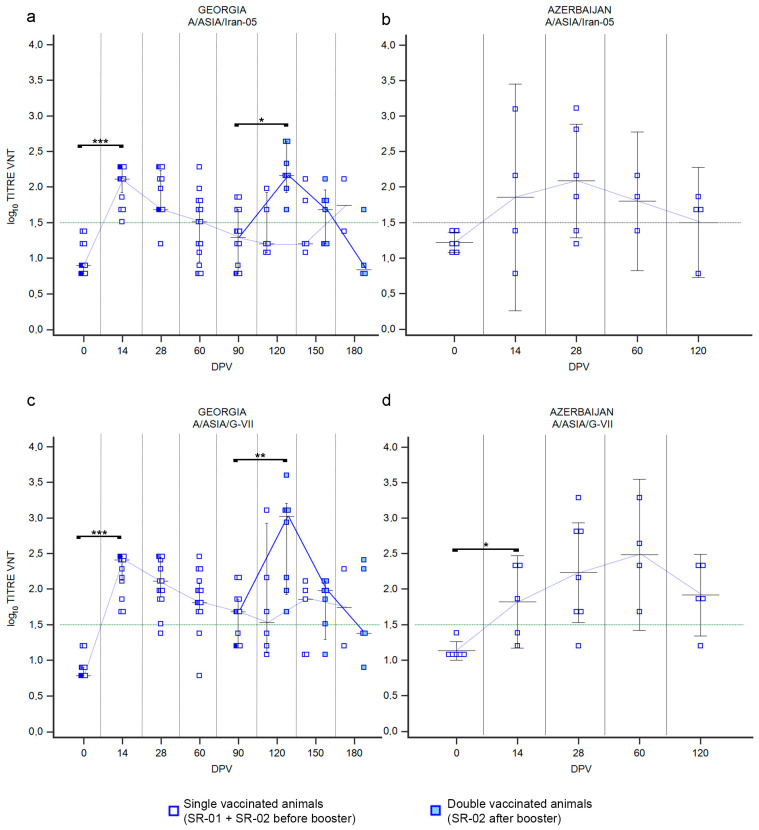
Immune response of small ruminants (SR) against FMDV serotype A. Charts showing the median, maximum and minimum log_10_ VNT titers obtained testing the neutralizing ability of SR sera against FMDV strain A/ASIA/Iran-05/TUR/06 (**a**,**b**) and strain A/ASIA/G-VII/NEP/84 (**c**,**d**). Sera from Georgia (**a**,**c**) and Azerbaijan (**b**,**d**). Abscissa, days post-vaccination (DPV); ordinate, log_10_ VNT titers. Error bar, 95% confidence interval; VNT positivity threshold, >1.5 log10 titer in green dashed line. Empty squares and dashed red line, single-vaccinated animals (SR-01 and SR-02 before booster); filled squares and solid red line, double-vaccinated animals (SR-02 group after booster). *** *p* value ≤ 0.001; **, *p* value ≤ 0.01; * *p* value ≤ 0.05.

**Figure 7 vaccines-12-00295-f007:**
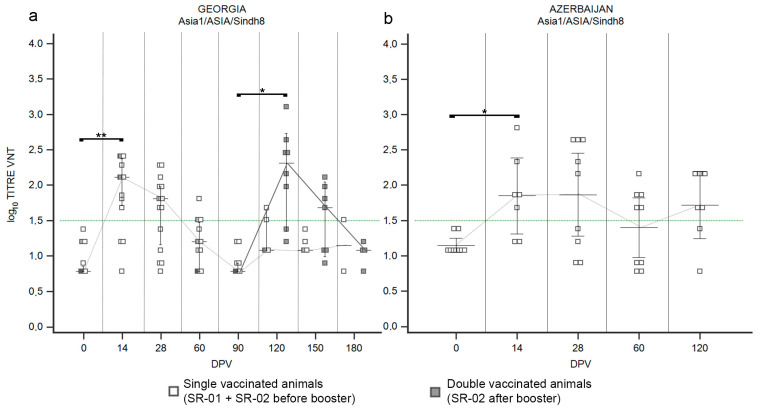
Immune response of small ruminants (SR) against FMDV serotype Asia1. Charts showing the median, maximum and minimum log_10_ VNT titers obtained testing the neutralizing ability of SR sera against FMDV strain Asia1/ASIA/Sindh8/TUR/15. Sera from Georgia (**a**) and Azerbaijan (**b**). Abscissa, days post-vaccination (DPV); ordinate, log_10_ VNT titers. Error bar, 95% confidence interval; VNT positivity threshold, >1.5 log10 titer in green dashed line. Empty squares and dashed grey line, single-vaccinated animals (SR-01 and SR-02 before booster); filled squares and solid grey line, double-vaccinated animals (SR-02 group after booster). **, *p* value ≤ 0.01; * *p* value ≤ 0.05 (*).

**Figure 8 vaccines-12-00295-f008:**
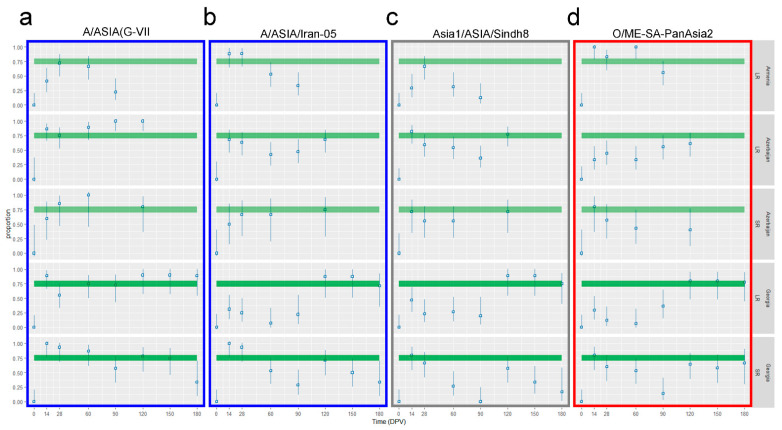
VNT positivity rate of vaccinated LR and SR. Charts showing the 95% confidence interval and point estimate of proportion animals with positive VNT titers against all the FMD strains included in the vaccine formula: A/ASIA/G-VII (**a**), A/ASIA/Iran-05 (**b**), Asia1/ASIA/Sindh8 (**c**) and O/ME-SA/PanAsia2 (**d**). The charts included all the ruminants and all three countries. From the top: LR from Armenia, LR from Azerbaijan, SR from Azerbaijan, LR from Georgia and SR from Georgia. Abscissa, time in days post-vaccination (DPV); ordinate, proportion of the population with VNT-positive titers (threshold of positivity > 1.5 log_10_ titer). The green areas highlight the proportion between 0.7 and 0.8 (70–80% of animals), which is the expected VNT-positive population rate for an effective vaccination.

**Table 1 vaccines-12-00295-t001:** List of sampling included in the analysis. Table present the list of the number of sera tested by the VNT and included in the analyses. The sera are grouped depending on: country of origin, type of ruminant (LR or SR), subgroup number (01–03), lineage used for the VNT and sampling time point.

			O/ME-SA/PanAsia 2								A/ASIA/Iran-05								A/ASIA/G-VII								Asia1/ASIA/Sindh8							
			0	14	28	60	90	120	150	180	0	14	28	60	90	120	150	180	0	14	28	60	90	120	150	180	0	14	28	60	90	120	150	180
** Georgia **	** LR **	**01**	8	8	8	4	2	1	1	-	8	8	8	4	2	1	1	-	9	9	9	4	2	1	1	-	9	9	9	4	2	1	1	-
		**02**	9	9	9	9	9	9	9	9	7	7	7	7	7	7	7	7	9	9	9	9	9	9	9	9	8	8	8	8	8	8	8	8
		**03**	2	2	2	2	2	2	1	2	2	2	2	2	2	2	1	2	2	2	2	2	2	2	1	2	2	2	2	2	2	2	1	2
	** SR **	**01**	9	6	6	6	5	5	4	2	9	6	6	6	5	5	4	2	9	6	6	6	5	5	4	2	9	6	6	6	5	5	4	2
		**02**	9	9	9	9	9	9	8	4	9	9	9	9	9	9	8	4	9	9	9	9	9	9	8	4	9	9	9	9	9	9	8	4
		**03**	2	2	2	2	2	2	2	-	2	2	2	2	2	2	2	-	2	2	2	2	2	2	2	-	2	2	2	2	2	2	2	-
** Azerbaijan **	** LR **	**01**	12	13	13	13	13	12	-	-	8	12	12	12	12	12	-	-	5	13	13	12	13	13	-	-	12	13	13	13	13	-	-	-
		**02**	6	6	6	6	6	6	-	-	2	6	6	5	6	6	-	-	2	9	7	7	8	9	-	-	9	9	9	9	9	-	-	-
		**03**	-	-	4	4	4	-	-	-	-	-	4	4	4	-	-	-	-	-	4	4	4	-	-	-	-	-	4	4	-	-	-	-
	** SR **	**01**	7	5	7	7	-	5	-	-	6	4	6	3	-	4	-	-	7	5	7	4	-	5	-	-	9	7	9	9	-	7	-	-
		**02**	-	-	-	-	-	-	-	-	-	-	-	-	-	-	-	-	-	-	-	-	-	-	-	-	-	-	-	-	-	-	-	-
		**03**	-	-	-	-	-	-	-	-	-	-	-	-	-	-	-	-	-	-	-	-	-	-	-	-	-	-	-	-	-	-	-	-
** Armenia **	** LR **	**01**	9	9	9	9	9	-	-	-	9	9	9	8	9	-	-	-	9	9	9	9	9	-	-	-	9	9	9	7	8	-	-	-
		**02**	9	7	9	9	9	-	-	-	9	7	9	9	9	-	-	-	9	7	9	9	9	-	-	-	9	7	9	9	8	-	-	-
		**03**	2	2	2	2	2	-	-	-	2	2	2	2	2	-	-	-	2	2	2	2	2	-	-	-	2	2	2	2	2	-	-	-

## Data Availability

The data presented in this study are available on request from the corresponding author. The data are not publicly available due to privacy restrictions.

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
