# Peer review of "Evaluation of Two Vaccines against Foot-and-Mouth Disease Used in Transcaucasian Countries by Small-Scale Immunogenicity Studies Conducted in Georgia, Azerbaijan and Armenia"

_vaccines, 2024, doi:10.3390/vaccines12030295_

Round 1

Reviewer 1 Report (Previous Reviewer 1)

Comments and Suggestions for Authors

I see that the "new" manuscript was improved for the first submission. I have a few comments:

Table 1, "Armenia" is not visible

Figure 2, for Georgia, I do not see in the data that the p value is less than 0.001 when comparing 0 and 14 dpv. The data for Azerbaijan only reach 120 days, why? The data for Armenia only reach 90 days, why? For these last data, a p value less than 0.001 is reported for 0 and 14 days, significant p values are also expected between 0 and 28 days and 0 and 60 days. For Armenia, you obtained what was established by the manufacturer, immunize every 90 days, right?

Please check the statistical analyses of all the figures. 

The conclusions establish what was already established by the manufacturers, boost ever 90 days. Moreover, the conclusions do not mention the high variability of the VNT results, they are all over the place. The conclusions should also mention that the VNT analyses should continue beyond 180 days to establish the true effectiveness of the vaccines evaluated.   

Comments on the Quality of English Language

Moderate editing must be applied. 

Author Response

Table 1, "Armenia" is not visible

The original Excel file we submitted with the draft was ok, and the name “Armenia” was correctly written and visible. This is an issue related to the Journal layout.

Figure 2, for Georgia, I do not see in the data that the p value is less than 0.001 when comparing 0 and 14 dpv.

The statistical analyses for each comparison were clearly reported in paragraph 2.9. Calculating the p value for the comparison between VNT titres against serotype O strain in Georgian LR time 0 and 14 DPV we obtained a p value of 0.001 using PRISM, so ≤ 0.001.

The data for Azerbaijan only reach 120 days, why? The data for Armenia only reach 90 days, why?

The discrepancies between the ideal vaccinating and sampling schemes and the real ones were clearly described by paragraph 2.5, Figure 1 and Table 1. In Azerbaijan Vet colleagues sampled animals only until 120 DPV while Armenian Vet colleagues until 90 DPV. I do not have specific information about the reason why it happened. Nevertheless, please remember that this kind of studies were conducted in countries in financial difficulties and based on the collaboration between governments and farmers, which sometimes face the urgency to sell or slaughter animals (grew calves for more than 3 months and avoid to sell them could be expensive). Moreover, please remember that social difficulties exist between Azerbaijan and Armenia, and that the Nagorno-Karabakh conflict interested the region between 1988 and 2024. The last two ideas are just speculations, but they should likely be the real (and untold) causes of those discrepancies.

For these last data, a p value less than 0.001 is reported for 0 and 14 days, significant p values are also expected between 0 and 28 days and 0 and 60 days.

As you can clearly notice (and we stressed it into the text) the VNT titres and the number of VNT positive animals decreased over the time. So, we did not expect that the difference between 0 and 28 or 60 days should be necessarily significant. We were interested in focusing our (and the readers’) attention on the difference in VNT situation after the vaccine administration, so on the couples 0-14 and 90-120. We calculated the p values for all the couples, but we reported in the figures only those important for our conclusions, in order to increase figures’ readability, as strongly recommended in previous review.

For Armenia, you obtained what was established by the manufacturer, immunize every 90 days, right?

As observed in the other cases, VNT titres and number of VNT positive animals decreased over the time in Armenia, too. For that reason, we speculated that also ARRIAH vaccine actually needs revaccination (lines 568-572). Not necessarily every 90 days. The assessment of the time between administrations was not the objective of the present work. The manufacturers provided that datum (90 days).

Please check the statistical analyses of all the figures. 

Thank you for the advice. We did it and everything seemed to be ok.

The conclusions establish what was already established by the manufacturers, boost ever 90 days.

Yes, but not exactly. As exhaustively described in the FAO/WOAH “Foot and mouth disease vaccination and post-vaccination monitoring” (reference #17) serological assays evaluating the immune response arose by vaccine administrations are very important for an independent preliminary assessment of a vaccine efficacy by the final users. In this case, data confirmed for example that animals need booster vaccinations, probably even closer than 90 days, but we do not have data to assess the period between one administration and the others (see above), reason why we corrected an error in line 612. The conclusion we reached are clearly summarized in lines 599-603, and of course in paragraph 5.  

Moreover, the conclusions do not mention the high variability of the VNT results, they are all over the place.

VNT is a complex technique. It needs high level biocontainment facilities, it is laborious, it requires trained staff and it does not show real repeatability features. For this reason, it is not so weird to obtain VNT titres with high variability, even in the same lab. It is a common acceptable and non-controversial feature of in-house serological methods, so an unnecessary information. It is clearly reported also in the FAO/WOAH “Foot and mouth disease vaccination and post-vaccination monitoring” (reference #17). Nevertheless, the variability of results is discussed in line 514-520, very briefly because in previous review the Revs asked us to make that part shorter.

The conclusions should also mention that the VNT analyses should continue beyond 180 days to establish the true effectiveness of the vaccines evaluated.  

We followed the FAO/WOAH guidelines (reference #17) that indicated for SSIS sampling until day 56 (we also added day 90 because it was the day of second vaccination) with an optional sampling six months after vaccination (180 DPV). The monitoring of vaccine coverage over the years in the population is not the topic of a SSIS but it might involve other kind of studies.

Reviewer 2 Report (Previous Reviewer 2)

Comments and Suggestions for Authors

I reviewed the manuscript entitled “Evaluation of two Vaccines against Foot-and-Mouth Disease used in Trans Caucasian Countries by Small-Scale Immunogenicity Studies conducted in Georgia, Azerbaijan and Armenia”. In this study authors conducted an evaluation of two FMDV vaccines under field conditions in three different countries.

Overall, I think this is an interesting study. However, under ideal conditions, the evaluation of two FMDV vaccines should be first conducted under experimental controlled conditions, preventing multiple variables including animal health condition and vaccine management. Also, challenge of vaccinated is an important part of the equation. Then, experiments in the field can be conducted.

These are some of my recommendations to improve the quality of this manuscript.

-Improve the discussion by adding more information about the epidemiological conditions of FMDV in these countries. By the time when this study was conducted, which FMDV strains were circulating in these countries. I suggest including a phylogenetic three showing the genetic relationship of the current viral strains circulating in these countries and the viral strains in the vaccine formulation.

-Authors state that during the experiment, no evidence of FMDV was recorded. Were animals tested during the experiment by the development of antibodies against non-structural proteins? Absence of infection was because vaccine efficacy? Or because absence of clinical cases around the zone where experiments were conducted. I think, a map showing the epidemiological situation of FMD around the places where these experiments were conducted, will put this study in a better perspective.

Author Response

However, under ideal conditions, the evaluation of two FMDV vaccines should be first conducted under experimental controlled conditions, preventing multiple variables including animal health condition and vaccine management. Also, challenge of vaccinated is an important part of the equation. Then, experiments in the field can be conducted.

Just a couple of clarification about this point. Obviously, the evaluation of a new vaccine could not start from a field study. The primary validation of a vaccine is usually conducted by the pharma manufacturing the vaccine itself. The procedure for the validation of vaccines against FMD follows very rigorous standard procedures and must to pass strict controls (see the European Pharmacopoeia 6.0, pages 918-920). The experiments described in this paper (which actually could not be properly defined field-experiments, because they were not PVM but trial conducted in farms under controlled conditions) were conducted after the routinely performed primary controls under experimental conditions, including the challenge for potency test which determined the PD50 of the vaccine. This kind of experiments under experimental conditions are usually conducted only on new vaccines (e.g. when the manufacturers change virus strains or industrial conditions of the production), while for new batches of well-known formulas those primary evaluation is not required. The European Pharmacopoeia 6.0 in paragraph 2.5.4 about FMD Vaccines elucidate the test necessary for the assessment of the potency of vaccine. Those are controls that can be carried out by the manufacturers, which have all the specific information about vaccine production. Indeed, the vaccines we evaluated had already passed manufacturers control before our tests. As reported in paragraph 2.3, we had already known the PD50 of the vaccine. The topic of the work was to evaluate the efficacy of the vaccines in inducing a good immune response, which means seropositivity in the VNT test involving at least 70% of the population. This kind of works are strongly recommended by WOAH (formerly known as OIE) and FAO. As exhaustively described in the FAO/WOAH “Foot and mouth disease vaccination and post-vaccination monitoring” (reference #17) serological assays evaluating the immune response arose by vaccine administrations are very important for an independent preliminary assessment of a vaccine efficacy by the final users. This is the reason why, starting from the few information of manufacturers, the final users (Vets from Georgia, Azerbaijan and Armenia) asked for an evaluation of effectiveness.

These are some of my recommendations to improve the quality of this manuscript.

-Improve the discussion by adding more information about the epidemiological conditions of FMDV in these countries. By the time when this study was conducted, which FMDV strains were circulating in these countries. I suggest including a phylogenetic three showing the genetic relationship of the current viral strains circulating in these countries and the viral strains in the vaccine formulation.

The epidemiological situation of the three countries involved in the work is already described in the Introduction (lines 47-59). The last incursion of FMD in TCC region happened in 2015 in Armenia, while the latest circulation events in Georgia and Azerbaijan are older (2011 and 2007). A brief description of the epidemiological situation of the countries neighbouring the TCC region is done in the introduction (lines 40-46), too. The details of epidemiology of FMD are easily available in WRLFMD Reports (references #3, #4 and #6). Moreover, we thought that this kind of information is not vital for the study. The topic of this study is not to evaluate the better formula (the most fitting strains to include in the vaccine) for that specific area, but to evaluate the efficacy (in terms of seroconversion) of the vaccine formula selected for field use by the Authorities. For the very same reason, a phylogenetic tree would not add any useful information about the assessment of the vaccine efficacy. Actually, manufacturers did not reveal the strains contained in the vaccine, but the lineages. And the study is based on strains of the very same lineages included in the formula. Thus, there are not the bases for a phylogenetic analysis at lineage level.

-Authors state that during the experiment, no evidence of FMDV was recorded. Were animals tested during the experiment by the development of antibodies against non-structural proteins?

Yes, we did it. Please see paragraph 2.6 and the remark of NSP ELISA negative results for LR and SR respectively line 228 and 360.

Absence of infection was because vaccine efficacy? Or because absence of clinical cases around the zone where experiments were conducted. I think, a map showing the epidemiological situation of FMD around the places where these experiments were conducted, will put this study in a better perspective.

As described for the first point, FMDV is not circulating in the countries since 2015. Negativity to NSP exclude the possibility of a novel incursion, and especially ensure that the detected immune responses arose exclusively because of the vaccination.

Round 2

Reviewer 2 Report (Previous Reviewer 2)

Comments and Suggestions for Authors

I thank the authors for their responses to my concerns. At this point, I don't have more questions about this study.

This manuscript is a resubmission of an earlier submission. The following is a list of the peer review reports and author responses from that submission.

Round 1

Reviewer 1 Report

Comments and Suggestions for Authors

These are my comments:

Minor:

Line 91, change "country" to "countries"

Line 132, erase "gravitational force"

Line 136-137, change " this field studies" to " these field studies" 

Line 453, " butefore"?

Line 466, change "antibody do not really" to "antibody does not really"

Line 476, " a little poorer"? 

Line 503, "and or"?

Line 506, check grammar in "to the low VNT titres arose immediately"

Major:

In the introduction section please include information (if available) regarding the economic impact of FMDV in cloven-hooved animals. How common the infection is? Are there other commercial vaccines available? Are there other vaccines that do not rely on inactivated FMDV?

In the discussion section, please comment on the current circulating strains per country (if data is available). Please also comment and the possibility of changing the vaccine formulation: increasing antigen content, increasing adjuvant content (saponin and alum), changing adjuvant, etc. Are there other studies that evaluate these vaccines? If so, please include them in the discussion section.  Finally, please comment of the wide range of results that you obtain in the log10 VNT titers, why this happens? Especially in Azerbaijani data.   

In the conclusions section erase all this: "As mentioned in the Materials and Methods (Vaccines, 2.3) both Shchelkovo Biocombinat and ARRIAH vaccines are comparable and this vaccine evaluation led to the following conclusions." Moreover, conclude on the necessity of changing the current vaccine formulations to have a long, sustained antibodies profile.

Comments on the Quality of English Language

English is fine, there are some minor typos and grammar errors though. 

Reviewer 2 Report

Comments and Suggestions for Authors

Overall, I consider that this study has several flaws in the methodology to support the results of this study. First, it is a complex study. Ideally the comparison between both vaccines should be performed under more controlled conditions. One of the main issues in this study is the potential presence of maternal antibodies in a good proportion of the animals, a situation that might have impaired the response of this animals to the vaccination. This situation is explained by the authors in the discussion, but I consider that is a main flaw in the study. 

Also, there are data in some of the experimental groups that is missing, so that affecting the comparisons performed in this study. In this sense, ideally, both vaccines should have been used in all countries to establish a fair comparison. 

Reviewer 3 Report

Comments and Suggestions for Authors

This study, as distilled in the abstract, is to assess homologous serum antibody titres to each of the components in the vaccine by way of VNTs. Lines 59-66 focus on the challenges related to vaccine matching which, whilst interesting, does not set the scene for why this study is undertaken. It is stated later that each of the vaccines has antigenic components assessed (assume as monovalent vaccines) as containing antigen equivalent to at least 6PD50. The background data on correlates of VNT with PD50 values should be referenced (in both monovalent and polyvalent preparations), which justifies the serological approach taken in this study. There should also be further reasoning on why there may a discrepancy.

Line 158-160: How homologous are these viruses with the components in the vaccine - references to support their use and justify a lack of significant antigenic differences are required as this is an essential part of the analysis. 

Line 162-163: The WOAH manual does not state as quoted in the paper. "Cut-off" values for "positivity" will vary depending on the objective. This paper appears to be testing the hypothesis that animals vaccinated with either of the 2 trivalent vaccines will seroconvert to a level in line with with the manufacturers label (equivalent to at least 6PD50). Therefore the cutoff value must be calibrated accordingly, according to the WOAH text:

"Cut-off titres for evaluating immunological protection afforded by vaccination have to be established from experience of potency test results with the relevant vaccine and target species."

If this is not the hypothesis, it must be clarified what the hypothesis is. 

Line 91-99 describing the expectation of study design regarding animal numbers and groups does not contain the statistical justification for this number (power calculations based on mean and SD of serological responses from similar studies etc), adds complexity to the paper as there were many deviations then described in supplementary table S1. Suggest the actual group numbers are listed in the main text in a clearer manner, not as a S1. Retrospective statistical analysis can be applied and results interpreted in this context. Within each of the 3 countries how were animals arranged, e.g. all LR on one epidemiological unit, SR on another, etc? Reason for LR-03 unvaccinated controls is not given, and these animals and associated results not mentioned again in the paper. Breeds and how ages were confirmed should be touched upon.

Line 115-134: sampling regimen should also be included in the same table as recommended above to enable the reader to understand. Should not be in S1 table as its important for reader to understand in main text. 

Line 164-172: the hypothesis being tested should be outlined, and justification for statistical method chosen to test this hypothesis explained. 

Line 168: Calves and Cattle are used interchangably within the report, recommend consistency and to use Cattle given the age range had already been specified. 

General point in results - the initial dilution in the VNT in WOAH is 1/4, and the cutoff should be given on the graphs as < x , x being the log10 of the first dilution tested. Graphs should have the countries below with abbreviated vaccines used in same tag for ease of review as data is spread between many graphs. Also, NSP negativity does not mean lack of exposure or even infection in vaccinated animals - the control animals would help to prove this?

Seropositive (NSP and VNT) animals should not be included in the results, and explained as such. Ideally these would have been screened before inclusion in the study, but understand logistics of field studies. 

The large ranges in titres for a given group and timepoint should be addressed and compared with what would be expected under controlled conditions - are human factors at play?

lines 194-205 (applicable subsequently): VNT results for positivity, as above, need to be defined. Positivity is not the same as cutoff predictive of protection, for example. Where Medians are significantly different to means this needs to be explained as normal distribution of immune responses would be expected. References to "situation remained stable" or "worsened" must be rephrased with reference to changes in titre only. 

line 447-478: Unclear by what is being referred to as "negative VNT titres. There was wide ranges of titres which are a cause for concern. Some formulations of FMD vaccines may induce good immune response for up to 6 months, however others require a boost after 1 - 2 months when starting the regimen. the QA testing at the factory will be a PD 50 conducted at 21days post vaccination, which this study seems to confirm, although the large variability in VNT , as mentioned, is a concern which should be addressed with reference to expected variation under controlled conditions. 

Lines 529-538. A more thorough literature review on duration of immune response to inactivated FMD vaccines should be undertaken as requirements for booster after 1-2 months is well known for some FMD vaccines, and could help contextualise the results. Given the wide variability in titres, a strong recommendation to describe in the conclusions would be to conduct such studies on duration of immunity / effect of boosters in a more controlled environment to help untangle any human factors in the variability.

Comments on the Quality of English Language

Good - minor revisions required as described in the text above and general clarification of terms, such as in table S1 which is confusing but which has been requested to be simplified and included as per the comments above.

Round 2

Reviewer 2 Report

Comments and Suggestions for Authors

...

Author Response

The solely comment of Reviewer 2 at the round 2 was that all the aspects of draft "must be improved".

Thanks to the very useful comments of Round 1 the text had already experimented a huge improvement and, without any other clear comments, the paper is in our opinion suitable to be considered for pubblication.

Regards